# Bee Assemblage in the Southern Chihuahuan Desert: The Role of Season, Year, and Trap Color in Abundance

**DOI:** 10.3390/insects14110875

**Published:** 2023-11-14

**Authors:** Esteban O. Munguia-Soto, Jordan Golubov, María C. Mandujano

**Affiliations:** 1Doctorado en Ciencias Biológicas y de la Salud, División de Ciencias Biológicas y de la Salud, Universidad Autónoma Metropolitana Xochimilco (UAM-X), Calzada del Hueso 1100, Colonia Villa Quietud, Coyoacán, Ciudad de Mexico 04960, Mexico; musesteban@gmail.com; 2Laboratorio de Ecología, Sistemática y Fisiología Vegetal, Departamento El Hombre y su Ambiente, Universidad Autónoma Metropolitana Xochimilco (UAM-X), Calzada Del Hueso 1100, Colonia Villa Quietud, Ciudad de Mexico 04960, Mexico; gfjordan@correo.xoc.uam.mx; 3Departamento Ecología de la Biodiversidad, Instituto de Ecología, Universidad Nacional Autónoma de México (UNAM), Circuito Exterior s/n, Ciudad Universitaria, Ciudad de Mexico 04510, Mexico

**Keywords:** conservation, declining, growth rate, pollinators, pollinators population solitary bees

## Abstract

**Simple Summary:**

There is increasing evidence of declining bee populations due to anthropogenic factors. We assessed the abundance of nine wild bee species over a four-year study period, estimating changes through monthly captures. *Apis mellifera*, the *Lasioglossum* (*Dialictus* spp.) complex, and *Macrotera sinaloana* had the largest population sizes and densities. Seasonal fluctuations were seen in most species in spring (March–May) coinciding with the flowering period of the main plant species and a low abundance during the winter months (December–January). Moreover, 77.7% of the populations showed a tendency to remain constant over time, with yellow tray traps capturing a higher number of individuals.

**Abstract:**

Recognizing how populations fluctuate over time is a crucial factor in determining the environmental elements affecting population persistence. However, the limited information on wild bee populations complicates the estimation of the impact of anthropogenic threats leading to changes in population size. To address this, we conducted a study capturing and monitoring nine species of wild bees through monthly samplings over four years. Tray traps were placed in permanent plots, and capture records were used to determine population size (*N*) and density (*D*). A generalized linear model (GLM) was employed to determine how the use of traps affected bee species captures. The families Apidae and Halictidae represented the most captures. *Apis mellifera*, the *Lasioglossum* (*Dialictus* spp.) complex, and *Macrotera sinaloana* exhibited the largest number of captures and highest population density. Most species (77.7%) showed a tendency to remain constant over the years and to have a higher number of captures in the spring months. Moreover, yellow traps were the most effective in capturing bee individuals. We suggest that the availability of essential resources and the reduction in environmental stressors positively affected the capture of wild bee populations.

## 1. Introduction

Bees are central for terrestrial ecosystems, as eighty-five to ninety percent of angiosperms depend on pollinator services [1,2]. There are close to 20,000 bee species worldwide, and the Mexican apifauna is regarded as one of the most diverse, encompassing approximately 144 genera and 1908 species across six families, namely Andrenidae, Apidae, Colletidae, Halictidae, Megachilidae, and Melittidae [3,4]. The highest bee richness is situated in the arid and semi-arid regions of North America, as well as those with a Mediterranean climate [3,5]. 

There is an internationally recognized pollinator crisis [6,7,8], with 40 percent of invertebrates having a threatened status and, on the IUCN, [9] red list; of these, 9% are bees, and bee populations have decreased by 37% [6]. This decrease in bee populations could have negative consequences for vegetation in terrestrial ecosystems and economic loss [10]. About 35% of global food crops are pollinated by these insects [6,7,10,11]. For example, the strong interrelationship between the cactus pear (*Opuntia* spp.) and solitary bees is critical for this staple crop in North America [12], and even species that are self-compatible, such as canola agroecosystems, require pollen exchange by *Apis mellifera* (Linnaeus, 1758) for optimal reproductive success [6,10]. The deterioration of natural environments, mostly due to anthropogenic activities such as agriculture, livestock husbandry, deforestation, and land use change, has led to a decrease in bee populations [13,14,15]. These activities have reduced patches of wild vegetation, leading to a loss of food (pollen and nectar), as well as resting, copulation, and nesting places, which are essential for the survival of bee populations [13,14,16] For example, Cane et al. [17] reported lower numbers of bee species and individuals in urbanized patches and surrounding vegetation than in wild vegetation (predominantly *Larrea tridentata* ((DC.) Coville, 1893), with species such as *Eucera venusta* (Timberlake, 1961), *Megandrena enceliae* (Cockerell, 1927), and *Perdita lateralis* (Timberlake, 1962) (ground-nesting or *L. tridentata* specialist bees) being nonexistent in urbanized patches.

The introduction of exotic species has further compounded the loss of bee populations [13]. Exotic species can be vectors of pathogens (e.g., *Varroa destructor* (Anderson & Trueman, 2000)), affecting bee populations in a relatively short period of time (between 2 and 4 years) [18]. They can also compete with native species, which results in a loss of native species (>10%) [19]. At the same time, the use of pesticides containing neonicotinoids causes physiological alterations to the nervous and reproductive system as well as immunosuppression, with negative impacts on bee populations [20]. While the decline in bee populations is acknowledged, studies assessing the decline in wild bee populations are scarce [21]. Most studies focus on economically important bees such as *Apis mellifera* and *Bombus* spp. [6,7], while few efforts have been made to assess the wild bees on which the vegetation of natural ecosystems depends.

Population size is one of the attributes most used by ecologists to determine the status of biological populations [22,23]. However, in practice, the estimation of the number of individual bees poses methodological problems, is often not as precise as desirable, and is generally measured with data from captured individuals to determine their abundance, frequency, and density [24]. Short-term studies of biological populations can provide information allowing us to quickly identify emerging patterns and potential trends of populations at a specific moment [25]. However, they must be complemented with long-term monitoring programs and other evaluation techniques to ensure the persistence of populations over time [25]. 

The aim of this study was to compare the population abundance and density of wild bee species over a four-year period to highlight potential trends, threats, and factors that favor bee populations in the southern Chihuahuan Desert.

## 2. Materials and Methods

### 2.1. Study Area

This study was carried out at the southernmost edge of the Chihuahuan Desert in the desert scrubland of Cadereyta de Montes, Querétaro, Mexico. The climate is hot and semi-arid with summer rains (BS1kw) [26], with an annual precipitation between 400 and 450 mm and an annual temperature that ranges between 16 and 18 °C [27]. These heterogeneous environmental conditions result in a varied mosaic of microphyllous, rosetophyllous, and carissicaulous scrub as the predominant vegetation [28]. Bee populations were sampled monthly for four years (2015, 2016, 2018, and 2019) in an area managed by the Cadereyta Regional Botanical Garden, “Ing. Manuel González de Cosío”. This 7.3 ha area maintains a conserved remnant of crassicaulous desert scrub, where the vegetation is characterized by mesquite (*Prosopis laevigata* (Humb. & Bonpl. Ex Willd.)), prickly pears (*Opuntia cantabrigiensis* (Lynch, 1903), *O. robusta* (H.L.Wendl. ex Pfeiff., 1837), *O. tomentosa* (Salm-Dyck, 1822), and *O. straptacantha* (Lem., 1839)), agave (*Agave mapisaga* (Trel., 1920)), leatherstem (*Jatropha dioica* (Cerv.)), fishhook cactus (*Mammillaria uncinata* (Zucc. Ex Pfeiff., 1837)), candy barrel cactus (*Ferocactus histrix* ((DC.) G.E.Linds., 1955)), blue myrtle cactus (*Myrtillocactus geometrizans* (Mart. Ex Pfeiff., 1897) Console), and yucca (*Yucca filifera* (Chabaud, 1876)) [29]. The area around the conserved space of the botanical garden is bounded by a *Y. filifera* plantation, an urbanized area, and an area with a remnant of xerophytic scrub that has been disturbed due to the introduction of livestock alongside the establishment of pastures for livestock feeding. Environmental variables such as mean temperature and precipitation and maximum wind speed (Table 1) were taken from the meteorological station found in the Botanical Garden [30]. 

### 2.2. Bee Capture

A three-day-long sampling period was carried out once per month for four years (2015, 2016, 2018, and 2019). Within the study site, 3 × 5 m plots were sampled in which three pan traps (yellow, blue, and purple for a total of 30 traps per sample period) were placed in each plot. Traps were placed on the ground and near flowering plants to capture the largest number of individuals [31]. Each pan trap (16 cm in diameter and 6.5 cm deep) had a 125 mL water solution with 5% commercial liquid soap [31]. These traps were distributed throughout all areas of the conserved area to encompass both central and peripheral sites. This arrangement sought to ensure comprehensive coverage of the various areas where bees occur [32]. The selection of colors has been shown to increase the diversity of species captured as they respond differently to color [33,34]. Furthermore, these colors match those of the flowers found in most plants in the study area. Traps were placed at 0900 h and removed at 2000 h every day, encompassing the opening hours of most of the flowers on the site [35]. Each color trap was 5 m apart since traps separated by smaller distances decreased the number of bees captured [35]. Collected bees were kept in 70% alcohol for later identification carried with the help of Dr. Ismael Hinojosa Diaz and Dr. Ricardo Ayala Barajas, Institute of Biology, UNAM.

### 2.3. Data Analysis

The abundance (N), the total number of individuals captured monthly as well as annually, their density (D = N/A) [36], and the number of individuals captured per unit of area (A = the area of the site is 7.3 ha) were recorded. We also evaluated the relationship between the total abundance of bees and environmental parameters for each sampling year using Pearson’s correlations [37]. Changes in abundance between sampled years were calculated using the cumulative abundance data for each month. The abundance between sampling periods reflects a relative change in abundance between time periods and would suggest constant captures over sampling periods or either declines or increases in the number of captured individuals. Finally, a generalized linear model (GLM) [38], suitable for abundance data, was used to analyze the relationship between abundance (dependent variable) and three factors: species, year, and color of the trap (independent variables).

## 3. Results

A total of 2208 individual bees were captured throughout the study period. These were classified into 12 species and the *Dialictus* complex, representing 12 genera and 4 families: Andrenidae (1 species), Apidae (6 species), Halictidae (5 species), and Megachilidae (3 species). *Ceratina* sp1 (*n* = 48), *Ceratina* sp2 (*n =* 33), *Agapostemon* sp. (*n* = 119), *Augochlora* sp. (*n* = 14), the *Lasioglossum* (*Evylaeus* spp.) complex (*n* = 63), and *Lithurgus planifrons* (*n* = 51) were only captured in 2015, an important indicator of absence. Eight bee species and the *Lasioglossum* (*Dialictus*) complex accounted for 1880 captured individuals found in at least two years (Table 2; Figure 1). The highest number of captures were for *A. mellifera* (*n* = 419), followed by *Lasioglossum* (*Dialictus* spp.) (*n* = 370) and *Macrotera sinaloana* (*n* = 347), while *B*. *pensylvanicus* (*n* = 21), *Melissodes* sp. (*n* = 63), and *Megachile* sp. (*n* = 88) had the lowest. Over time, the abundance of some captured species remained relatively constant (Figure 1), while that of others (*A*. *mellifera* and *B*. *pensylvanicus*) decreased (Figure 1). The *A. mellifera* population in the study site is not managed and possibly escaped from a bee farm located in the region. No captures of individuals of *B. pensylvanicus* were obtained after 2016, while *Melissodes* sp. and *Lithurgus littoralis* were absent in 2015 (Table 2; Figure 1). Regarding the average density over the years, the species that obtained more than 10 ind • ha^−1^ were *M*. *sinaloana*, *A. mellifera*, and the *Lasioglossum* (*Dialictus* spp.) complex (Table 2), while the rest had a low average density (<10 ind • ha^−1^).

The correlation between abundance and the environmental variables (Table 1) showed that over the four years, all three environmental variables had a positive relationship with abundance; however, the correlation was not always significant. Temperature had the highest correlation values during the four years (Table 1), with 2016 and 2019 having the highest correlation values. On the other hand, wind speed in 2015 had a negative correlation with bee abundance. 

There was an increase in abundance within years over the course of the sampling period (Figure 2). Spring (March, April, and May) clearly reflected increased activity of solitary bee populations, as flying adults emerged from nests. Halfway through the year, there was a gradual decrease in the number of individuals for many species, such as *Melissodes* sp., of which we have captured no individuals as of August, or *Megachile* sp., which we stopped recording in the month of October. In December, only individuals of *Augochlorella* sp., *B. pensylvanicus*, the *Lasioglossum* (*Dialictus* sp.) complex, and *M*. *sinaloana* were captured. *A*. *mellifera* was present throughout the year, being especially abundant in the latter half (Figure 2). Lastly, the GLM analysis revealed that bee species (χ^2^ = 890.69, *p* < 2 × 10^−16^, df = 8), pan trap color (χ^2^ = 571.43, *p* < 2 × 10^−16^, df = 2), year (χ^2^ = 17.02, *p* < 0.0007, df = 3), and year x species interactions were significant (χ^2^ = 202.26, *p* < 2 × 10^−16^, df = 24). Even though the year was important, abundance was only significantly different between 2018 and 2019 (*p* = 0.0003). The yellow trap (*n* = 1049) captured the highest number of individuals, followed by the blue trap (*n* = 602), and, lastly, the violet trap (*n* = 229). Most captured individuals belonged to the *Lasioglossum* (*Dialictus* spp.) complex. This suggests that both the color of the trap and the bee species are influential factors in captures in the study area. However, the year x color (χ^2^ = 4.84, *p* < 0.56, df = 6) but not species × color interactions were significant (χ^2^ = 25.49, *p* < 0.06, df = 16). 

## 4. Discussion

Our study revealed that of the eight species and one species complex caught over a span greater than one year, seven bee species experienced fluctuations in their abundance and density within each year with a tendency to remain relatively constant over time, while one exhibited a decline, and one showed an increase. This tendency to remain constant could be attributed to the availability of between-year resources and stability of environmental factors such as temperature, precipitation, and wind speed in the study area, which is consistent with previous studies [4,5,39]. Most individuals were captured during the months of March to May across the four sampled years, a time when the flowering peaks of cacti and the other dominant species are seen, suggesting that most bee populations have life cycles that are synchronized with the flowering of the plants during that period. This observation is inconsistent with Roubik [40], who found a change in the relationships between bees and the vegetation of a tropical forest due to a delay in flowering and the availability of resources. We found that bee abundance was correlated with annual temperature, with the optimal temperature ranging between 20 to 35 °C, consistent with that reported by Ortiz-Sánchez and Aguirre-Segura [39]. There was little variation between years for most species; however, changing environmental conditions over a longer period may alter these conditions. This was clearly seen with the absence of five species and the *Lasioglossum* (*Evylaeus*) complex after the 2015 sampling period.

Species were most abundant in the months of March, April, and May, declining after June. Most of these bee species are solitary, with univoltine life cycles (species that reproduce only once a year and have a diapause during the larval phase, which in this case can last for a year). The changes in seasonal abundance were related to the nature of the life cycle, with the new cohort of flying adults emerging the following year [5]. Many plants species flower between March and September in the Chihuahuan Desert, with spring being the time when there is the highest production of floral resources [41]. This synchronization with floral resources was also observed by Minckley et al. [42], who identified the spatio–temporal interaction between bees and their floral resources, especially with *Larrea tridentata*, in the Chihuahuan, Sonoran, and Mojave Deserts. In contrast, three species, *Apis mellifera*, *Macrotera sinaloana*, *Diadasia* sp., and the *Lasioglossum* (*Dialictus* sp.) complex, were found year-round and during the four-year study period. The native solitary species (*Macrotera sinaloana*, *Lasioglossum* (*Dialictus* sp.) complex, and *Diadasia* sp.) have bivoltine life cycles (two generations per year); however, these species also undergo diapause in both the larval and adult stages, which helps them maintain their populations over time [5]. Packer et al. [43] reported that several species of Halictidae, among them *Lasioglossum* (*Dialictus* sp.) *laevissimum*, are bivoltine, with the first individuals emerging in June and a second generation of individuals in August. Neff and Simpson [44] also mention *Diadasia rinconis* as having two generations per year, one that appears when there is an abundance of flowers of three species of *Opuntia* (*O*. *leptocaulis*, *O*. *macrorhiza*, and *O*. *engelmannii* var. *lindheimeri*) and the other long after the peak of flowering; this second generation generally dies without reproducing.

Indirect capture methods, such as the use of pan traps, have limitations in estimating the abundance of bee populations [45] and may not reflect the actual bee diversity and population dynamics. These limitations include bias in the capture of individuals due to the lack of representativeness of the sample with respect to the total population, mortality of individuals during the study, environmental factors at the time of capture, and the color of the trap [45,46]. Pan traps have also been shown to favor the capture of Halictids [47], as found in this study. In contrast, direct capture methods, such as marking and recapturing, provide more accurate estimates of abundance since they allow for a representative sample of the total population, as well as good estimates of demographic parameters such as survival and mortality [46,48,49]. However, the latter methods are expensive and labor-intensive. Nevertheless, despite the limitations of indirect capture methods, they have certain advantages over direct methods: they allow for obtaining information on the distribution and abundance of populations in limited areas (such as solitary bees), avoiding overestimation of abundance, and underestimation of mortality [49]. To minimize capture bias, our study was carried out in a specific area where relevant aspects, such as life histories, feeding (oligolectic or polylectic), nesting, and flight ranges of bee populations were considered to properly place traps [5,50]. 

We found that 77% of bee abundances were rather constant over the four-year study period, while 11% increased. This stability may be due to the conditions surrounding the botanical garden, which potentially provide the necessary resources to maintain the populations [5,50]. It is important to note that short-term studies have limitations and may not provide a complete picture of wild bee populations at the study site; however, with these results, we can suggest that a management plan that includes the relationship between bee populations and plant species is needed. Plants such as prickly pears (*Opuntias* sp.), mesquite (*Prosopis laevigata*), and small shrubs and herbs (*Stevia* sp., *Asclepias* sp., *Sphaeralcea* sp., and *Mimosa* sp.) can promote the stability of bee populations and are able to conserve the necessary habitat conditions, such as soil characteristics and the presence of adequate microhabitats for the survival of the community [51,52]. Conservation policies are usually directed more towards the networks of species that interact in the ecosystem on a large scale than concentrating conservation efforts on a single species and its specific habitat [53]. However, in some cases, the management of an ecosystem can be guided by the need of a particular species or taxon, such that bees can function as a group of flagship species for plants and other animals in arid ecosystems given their importance and the richness found in these environments [50].

## 5. Conclusions

Although indirect sampling methods and short-term studies have their limitations, these methods can aid in providing preliminary information and are cost-effective and accessible means of assessing abundance within terrestrial ecosystems. This study highlights the importance of pan trap color, year, season, and species for bee abundance in the southern Chihuahuan Desert.

## Figures and Tables

**Figure 1 insects-14-00875-f001:**
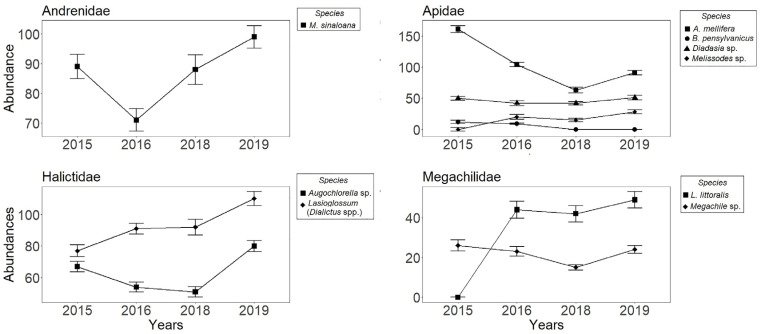
Abundance of bees (±SE) collected in pan traps and grouped by family in Cadereyta de Montes, Querétaro, México. Data from monthly sampling across four years: 2015, 2016, 2018, and 2019.

**Figure 2 insects-14-00875-f002:**
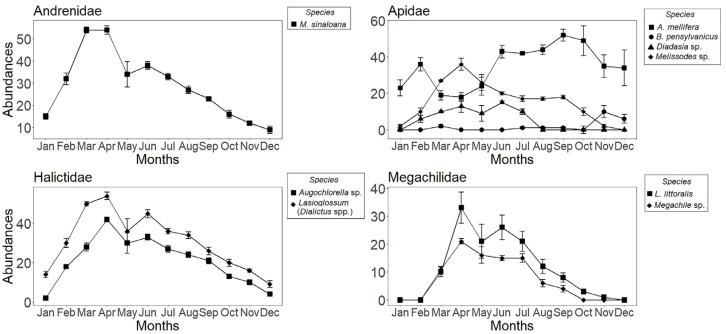
Abundance of bees (±SE) per month of capture in Cadereyta de Montes, Querétaro, Mexico. Species are shown by family. Data from data monthly sampling across four years: 2015, 2016, 2018 and 2019.

**Table 1 insects-14-00875-t001:** Average and standard deviation of the environmental parameters, as well as the Pearson regression coefficients and the estimated *p*-values between the total abundance of bees with each environmental parameter (temperature, precipitation, and wind speed). Data from monthly sampling of wild bees across four years (2015, 2016, 2018, and 2019), Cadereyta de Montes, Querétaro, Mexico. Environmental parameters are from the meteorological station at the Botanical Garden [30].

Years	Temperature (°C)				Precipitation (mm)				Wind Speed (km/h)			
	Average	sd	R	*p*-Value	Average	sd	R	*p*-Value	Average	sd	R	*p*-Value
2015	16.1	2.17	0.65	0.02	24.3	30.16	0.42	0.17	16.9	3.44	−0.15	0.63
2016	16.4	2.92	0.82	0.001	33.3	34.31	0.19	0.54	16.7	1.86	0.58	0.05
2018	16.8	2.84	0.57	0.05	17.9	34.9	0.42	0.18	16.9	2.31	0.3	0.34
2019	17.8	2.64	0.83	0.001	17.1	22.14	0.16	0.62	16.1	1.8	0.55	0.07

**Table 2 insects-14-00875-t002:** Families and species of bees present in Cadereyta de Montes, Querétaro, Mexico. Abundance (N) and density (D, ind • ha^−1^) for each year of sampling, total abundances, and average density over the study period (monthly sampling across four years: 2015, 2016, 2018, and 2019).

Family	Genus and Species	*N* _15_	$\bar{\chi }$ _15_	*σ* _15_	*D* _15_	*N* _16_	$\bar{\chi }$ _16_	*σ* _16_	*D* _16_	*N* _18_	$\bar{\chi }$ _18_	*σ* _18_	*D* _18_	*N* _19_	$\bar{\chi }$ _19_	*σ* _19_	*D* _2019_	*N_total_*	$D\bar{\chi }$
Andrenidae	*Macrotera sinaloana* (Timberlake, 1958)	89	7.42	4.08	12.71	71	5.92	3.78	10.14	88	7.33	4.98	12.57	99	8.25	3.79	14.14	347	12.39
Apidae	*Apis mellifera* (Linnaeus, 1758)	161	13.42	5.38	23.00	104	8.67	3.60	14.86	63	5.25	4.54	9.00	91	7.58	3.85	13.00	419	14.96
	*Bombus pensylvanicus* (De Geer, 1773)	12	1.00	2.37	1.71	9	0.75	0.97	1.29	0	0.00	0.00	0.00	0	0.00	0.00	0.00	21	0.75
	*Diadasia* sp.	50	4.17	2.98	7.14	42	3.50	3.92	6.00	42	3.50	2.71	6.00	51	4.25	3.33	7.29	185	6.61
	*Melissodes* sp.	0	0.00	0.00	0.00	20	1.67	1.92	2.86	15	1.25	1.96	2.14	28	2.33	2.50	4.00	63	2.25
Halictidae	*Augochlorella* sp.	67	5.58	3.37	9.57	54	4.50	3.18	7.71	51	4.25	3.36	7.29	80	6.67	3.55	11.43	252	9.00
	*Lasioglossum* (*Dialictus* spp.) complex	77	6.42	3.63	11.00	91	7.58	3.32	13.00	92	7.67	4.96	13.14	110	9.17	4.39	15.71	370	13.21
Megachilidae	*Lithurgus littoralis* (Cockerell, 1917)	0	0.00	0.00	0.00	44	3.67	4.31	6.29	42	3.50	4.15	6.00	49	4.08	4.06	7.00	135	4.82
	*Megachile* sp.	26	2.17	2.82	3.56	23	1.92	2.39	3.29	15	1.25	1.42	2.14	24	2.00	1.91	3.43	88	3.10

## Data Availability

The databases used are available in the repository https://github.com/Esteban-Mun10/Mungu-a_etal_bees (accessed on 10 October 2023).

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
