# Peer review of "Bee Assemblage in the Southern Chihuahuan Desert: The Role of Season, Year, and Trap Color in Abundance"

_insects, 2023, doi:10.3390/insects14110875_

Round 1
Reviewer 1 Report
Comments and Suggestions for Authors
Greetings! Thank you for this paper. It is an interesting study with interesting results. I think there should be a revision of this manuscript. I don’t think the revision needs to be extensive, but I think there are some missing results or general information about the study or discussion points that should be incorporated in the manuscript. I comment on these things below. There are also a few areas of very confusing writing. I point these areas out as well.
Finally, I have a concern about ALL the tables and figures. I think the margins are not properly set and these will run off the page. The figures are way too small- I cannot read the legends. Make sure all captions are on the same page as the figure/table. There are spacing issues throughout the paper. Those need to be corrected. (I’m reading a printed version of the manuscript I downloaded from the publisher.)
Introduction
Please discuss why you think 4 years is enough time to calculate a meaningful population growth rate.
Are there any past collections that allow a comparison of species present and their relative abundance? Please add a comment on this. Maybe this is the first time anyone has studied the bee community of this garden?
Materials and Methods
In this section, please add a short description of the surrounding landscape and the suitability of this landscape for bee survival.
Did you place traps in the center of the 7.3 ha site, or near edges? Were the traps within flight distance of most solitary bees to areas outside the study site? Important information for assessing your results.
I would justify the purple pan. Most pan trapping uses yellow, blue and white. Why purple? Did you see a difference in the tendency of a color to trap either a higher abundance than the other color traps or to trap a species with a greater tendency?
I seriously question using your data to generate a species status. Many bee species are found in surveys in extremely low numbers or only once in consecutive years. Can you really assess the status of a species based on your small study at such a small site? You can say whether or not it is common in your study site and try to quantify this abundance or talk about the population trend seen over the four years, but to assess its status based on your study? That I find hard to believe. Unless you can justify this quite firmly, leave out the species status and just talk about declining/increasing or steady occurrence of the species. When you go the species status level, then you make unsupportable claims (as in line 247) that rare species are EW.
Results
Line 184: You must be more clear about your Lasioglossum (Dialictus) category that you are treating as one species. Are you lumping all the Dialictus into one species? I have a very hard time believing that all 370 of those bees were one species. How many species or morpho-species do you actually have there? You should re-run the analyses with a corrected number of species in the Lasioglossum genus, or justify not doing so. (On the other hand, you can probably consider every taxon at the genus level only. Consider this for the analysis.).
With the analyses: I’m a little concerned with leaving the Apis in the analyses. Given the size of the study site, these could easily be bees from surrounding farms and urban areas. If that is the case, of course they are found in stable abundance. You should discuss this, or run analyses with and without Apis and see how the results compare. I think this important with the Pearson correlations with bee abundance and environmental conditions. Are there so many Apis present in your dataset that they are driving the correlations?
Line 195: spacing problem
Paragraph starting at line 201: the correlation is not always strong or significant. This should be pointed out.
Line 207: spacing problem
Line 212: the word should be “captured”
Paragraph starting line 217: The entire paragraph needs re-writing for clarity.
Discussion
Please revise the opening sentence (line 249) for clarity. I suggest, “Our study revealed that of the 9 species caught in more than one year…”
The discussion will need adjustment based on the changes in analysis mentioned above.
References
Please revise the references. In the text, I cannot find reference 17 or 64. The mention of references 28 and 29 are inverted (i.e. 29 comes before 28).
Thank you!
Comments on the Quality of English Language
Fine
Author Response
- The title was changed such that it now focuses on bee abundance and factors that affect it.
- También se eliminó la tabla 3 y la información complementaria "S1".
Introduction
Please discuss why you think 4 years is enough time to calculate a meaningful population growth rate.
R= A justification for why short-term samplings is good was added in lines 79-81. Even though long term studies are more meaningful, we have changed the emphasis of the paper to focus on changes in abundance more than estimation of population sizes as suggested by reviewer 2.
Are there any past collections that allow a comparison of species present and their relative abundance? Please add a comment on this. Maybe this is the first time anyone has studied the bee community of this garden?
R= There is no previous species list recorded for the area so it is difficult to compare data. This would be the first baseline dataset for the area.
Materials and Methods
In this section, please add a short description of the surrounding landscape and the suitability of this landscape for bee survival.
R= Added a short description of the surrounding area in lines 102-107. We also discuss the importance of the area to maintain bee abundance over time
Did you place traps in the center of the 7.3 ha site, or near edges? Were the traps within flight distance of most solitary bees to areas outside the study site? Important information for assessing your results.
R= we describe the areas of the study site where the traps were placed, they encompassed areas within the whole 7.3 ha (center as well as edges). These are described in lines 120-122
I would justify the purple pan. Most pan trapping uses yellow, blue and white. Why purple? Did you see a difference in the tendency of a color to trap either a higher abundance than the other color traps or to trap a species with a greater tendency?
R= A justification was added as to why the colors yellow, blue and violet were used in the study site in lines 123-125. There are a several plant species in the study site that have violet flowers (Echinocereus cinerascens, E. pentalophus, Ipomoea cappilacea and I. luzanii, Mimosa aculaticarpa, Sphaeralcea angustifolia, Anoda cristata, Bouchea primatica among others). We also tested the effect of color in the analysis and even though the violet pan traps caught the least amount of bees they still contributed close to 12% of total captures
I seriously question using your data to generate a species status. Many bee species are found in surveys in extremely low numbers or only once in consecutive years. Can you really assess the status of a species based on your small study at such a small site? You can say whether or not it is common in your study site and try to quantify this abundance or talk about the population trend seen over the four years, but to assess its status based on your study? That I find hard to believe. Unless you can justify this quite firmly, leave out the species status and just talk about declining/increasing or steady occurrence of the species. When you go the species status level, then you make unsupportable claims (as in line 247) that rare species are EW.
R= A second reviewer also questioned the use of pan traps to assess population status so the entire methodology and focus of the manuscript changed. The sections that mentioned population growth rates and bee status were removed, emphasizing changes in abundance ane the effecto of color, season and year on capture frequencies. An analysis was added to describe the influence of pan trap color, year and species on the abundance of bee species.
Results
Line 184: You must be more clear about your Lasioglossum (Dialictus) category that you are treating as one species. Are you lumping all the Dialictus into one species? I have a very hard time believing that all 370 of those bees were one species. How many species or morpho-species do you actually have there? You should re-run the analyses with a corrected number of species in the Lasioglossum genus, or justify not doing so. (On the other hand, you can probably consider every taxon at the genus level only. Consider this for the analysis.).
R= For the case of Lasioglossum (Dialictus), we followed the suggestion to keep it as a genus and use the term Lasioglossum (Dialictus spp) complex. Unfortunately this group is largely unknown and no clear taxonomic treatments have been done. As the emphasis is now on abundance data more than population size and change we feel the, the need to identify species becomes less relevant.
With the analyses: I’m a little concerned with leaving the Apis in the analyses. Given the size of the study site, these could easily be bees from surrounding farms and urban areas. If that is the case, of course they are found in stable abundance. You should discuss this, or run analyses with and without Apis and see how the results compare. I think this important with the Pearson correlations with bee abundance and environmental conditions. Are there so many Apis present in your dataset that they are driving the correlations?
Line 195: spacing problem
R= Corrected
Paragraph starting at line 201: the correlation is not always strong or significant. This should be pointed out.
R= It has already been noted that the correlations were not always significant
Line 207: spacing problem
R= Corrected
Line 212: the word should be “captured”
R= Corrected
Paragraph starting line 217: The entire paragraph needs re-writing for clarity.
R= This entire paragraph was eliminated.
Discussion
Please revise the opening sentence (line 249) for clarity. I suggest, “Our study revealed that of the 9 species caught in more than one year…”
This sentence was changed as suggested
The discussion will need adjustment based on the changes in analysis mentioned above.
The discussion was changed leaving out any reference to population status and population size or trends. We now have an emphasis on the factors (pan trap color, year and season) that affected the abundance of bees species captured.
References
Please revise the references. In the text, I cannot find reference 17 or 64. The mention of references 28 and 29 are inverted (i.e. 29 comes before 28).
R= Corrected
Thank you!
Reviewer 2 Report
Comments and Suggestions for Authors
While this paper presents a very important research question and clearly describes the background and results, I believe the research methods are fundamentally flawed in a way that makes this paper not suitable for publication, unless substantial adjustments are made to how the data are discussed. Pan trap captures cannot be used to estimate population sizes and population growth rates of bee populations. This is supported by a many published papers, usefully summarized by Portman et al. in a 2020 paper (Portman et al. 2020. The state of bee monitoring in the United States: A call to refocus away from bowl traps and toward more effective methods. Annals of the Entomological Society of America 113: 337-342). The authors could still draw inference on the bee species presented in their paper with a careful reframing of the paper away from population size and growth to discuss the rate of capture of the species only. This may suggest something about their populations, but cannot be used to draw conclusions about demography and possible declines.
Author Response
- The title was changed such that it now focuses on bee abundance and factors that affect it.
- También se eliminó la tabla 3 y la información complementaria "S1".
While this paper presents a very important research question and clearly describes the background and results, I believe the research methods are fundamentally flawed in a way that makes this paper not suitable for publication, unless substantial adjustments are made to how the data are discussed. Pan trap captures cannot be used to estimate population sizes and population growth rates of bee populations. This is supported by a many published papers, usefully summarized by Portman et al. in a 2020 paper (Portman et al. 2020. The state of bee monitoring in the United States: A call to refocus away from bowl traps and toward more effective methods. Annals of the Entomological Society of America 113: 337-342). The authors could still draw inference on the bee species presented in their paper with a careful reframing of the paper away from population size and growth to discuss the rate of capture of the species only. This may suggest something about their populations but cannot be used to draw conclusions about demography and possible declines.
R= Reviewer 1 also raised the same issue and we agree. We have now changed the emphasis of the paper away from population status and size and more towards the relative abundance and the effect of year, season, and pan trap color on bee abundance. We do feel four years of data can provide some inkling as to what species are usually found, especially the halictids. We hope this new version that steers away from the problems raised of estimating population size with pan traps is more adequate.
Reviewer 3 Report
Comments and Suggestions for Authors
Corrections and suggestions are made in the text.
For the appendix: should references cited be included in the references section of the text?

Author Response
General changes
- The title was changed such that it now focuses on bee abundance and factors that affect it.
- También se eliminó la tabla 3 y la información complementaria "S1".
Corrections and suggestions are made in the text.
Abstract:
Add "sp."
R= “spp” was added to all Lasioglossum (Dialictus) complex.
Regions with Mediterranean climate are among the richer as well (Michener, 2007)
R= We added that Mediterranean climates also have a lot of species richness, line 43
Since this point, species names appear and they has to be mentioned, at least the first time, with author and year of description
R= The year of description and author was added to all species
in italics
R= Corrected in Apis mellifera
I do not find the reference [17]
R= Reference 17 was placed, line 58
This seems to mean that the authors are generalizing but it is not always true
R= The beginning of this paragraph was modified so as not to generalize that all authors say this idea, line 64
Materials and Method
I think it has to be changed to "[34]"
R= The citation was corrected
Do you mean "[15]" or "[35]"?
R= The citation was changed to the correct one, which is now citation 31
As far as I know, Michener (2007) has no specific keys for this study. Probably, the authors have used them just to reach the family or genus level, but they would have to specify if they have used reference collections or if the specific identification was fully carried out by Dr Hinojosa
R= This paragraph is modified saying that Dr. Ismael Hinojosa and Dr. Ricardo Ayala helped identify the captured species, lines 129-130
This phrase and the application of this criterion are false. The authors can not base their conclusions on data not obtained by themselves in this particular experimentation.
R= This idea was removed from the text
And now, what is stated here does not match with what the authors say in the end of the previous paragraph
R= That paragraph was removed
Results
add "sp.”
R= “spp” was added to all references to the Lasioglossum (Dialictus) complex
deete "solitary" because not all species are solitary
R= The word 'solitary' was removed from all tables and figures
separate "each year"
R= The words were separated every year. Line 165
change to "S1"?
R= This reference was removed as well as the supplementary figure
In this point I has to say that I do not think that the criteria used by IUCN for the categories may be used here for such small territory studied in this research. Moreover, the authors do not use the criteria in the same way that are used for catalogue species at a national or regional level. This analysis should be eliminated. Besides, the study covers a very short lapse of time and conclusions cannot be definitive.
R= That paragraph and the analysis regarding IUCN status as well as population trends and sizes was removed from the document
Apis mellifera, at this geographic range level, cannot be categorized as EN because its populations depend on the beekeeping in the region.
R= The risk categories were removed
Discussion
add "sp."
R= “spp” was added to all references to the Lasioglossum (Dialictus) complex.
I do not find the reference "[64]"
R= The entire bibliography was corrected, and there are now only 56 bibliographic citations
References
this reference is cited twice
R= The second reference was removed, line 311
"Apoidea" (capital initial letter)
R= In reference 10, 'Apoidea' was capitalized.
Title complete, please
R= In reference 22, the name of the reference was added.
name complete
R= In citation 29, the full name of the author was added
the journal name is "Eos"
R= In citation 39, the name of the journal “Eos” was changed
Round 2
Reviewer 2 Report
Comments and Suggestions for Authors
I applaud the authors for changing the emphasis and scope of the paper to address questions that are more appropriate to the methods they employed (pan traps).
Reviewer 3 Report
Comments and Suggestions for Authors
The MS is now corrected and can be accepted for publication.